# Distribution, Ecology, Chemistry and Toxicology of Plant Stinging Hairs

**DOI:** 10.3390/toxins13020141

**Published:** 2021-02-13

**Authors:** Hans-Jürgen Ensikat, Hannah Wessely, Marianne Engeser, Maximilian Weigend

**Affiliations:** 1Nees-Institut für Biodiversität der Pflanzen, Universität Bonn, 53115 Bonn, Germany; ensikat@unitybox.de; 2Kekulé-Institut für Organische Chemie und Biochemie, Universität Bonn, Gerhard-Domagk-Str. 1, 53129 Bonn, Germany; Hannah.Wessely@uni-bonn.de (H.W.); Marianne.Engeser@uni-bonn.de (M.E.)

**Keywords:** *Urtica*, *Dendrocnide*, *Tragia*, Loasaceae, neurotransmitters, acetylcholine, histamine, herbivores, defense mechanisms, toxicity

## Abstract

Plant stinging hairs have fascinated humans for time immemorial. True stinging hairs are highly specialized plant structures that are able to inject a physiologically active liquid into the skin and can be differentiated from irritant hairs (causing mechanical damage only). Stinging hairs can be classified into two basic types: *Urtica*-type stinging hairs with the classical “hypodermic syringe” mechanism expelling only liquid, and *Tragia*-type stinging hairs expelling a liquid together with a sharp crystal. In total, there are some 650 plant species with stinging hairs across five remotely related plant families (i.e., belonging to different plant orders). The family Urticaceae (order Rosales) includes a total of ca. 150 stinging representatives, amongst them the well-known stinging nettles (genus *Urtica*). There are also some 200 stinging species in Loasaceae (order Cornales), ca. 250 stinging species in Euphorbiaceae (order Malphigiales), a handful of species in Namaceae (order Boraginales), and one in Caricaceae (order Brassicales). Stinging hairs are commonly found on most aerial parts of the plants, especially the stem and leaves, but sometimes also on flowers and fruits. The ecological role of stinging hairs in plants seems to be essentially defense against mammalian herbivores, while they appear to be essentially inefficient against invertebrate pests. Stinging plants are therefore frequent pasture weeds across different taxa and geographical zones. Stinging hairs are usually combined with additional chemical and/or mechanical defenses in plants and are not a standalone mechanism. The physiological effects of stinging hairs on humans vary widely between stinging plants and range from a slight itch, skin rash (urticaria), and oedema to sharp pain and even serious neurological disorders such as neuropathy. Numerous studies have attempted to elucidate the chemical basis of the physiological effects. Since the middle of the 20th century, neurotransmitters (acetylcholine, histamine, serotonin) have been repeatedly detected in stinging hairs of Urticaceae, but recent analyses of Loasaceae stinging hair fluids revealed high variability in their composition and content of neurotransmitters. These substances can explain some of the physiological effects of stinging hairs, but fail to completely explain neuropathic effects, pointing to some yet unidentified neurotoxin. Inorganic ions (e.g., potassium) are detected in stinging hairs and could have synergistic effects. Very recently, ultrastable miniproteins dubbed “gympietides” have been reported from two species of *Dendrocnide*, arguably the most violently stinging plant. Gympietides are shown to be highly neurotoxic, providing a convincing explanation for *Dendrocnide* toxicity. For the roughly 648 remaining stinging plant species, similarly convincing data on toxicity are still lacking.

## 1. Introduction 

Stinging hairs of the common stinging nettle (*Urtica dioica* L.) and other nettles have fascinated botanists for centuries and have been studied extensively [1]. Indeed, stinging hairs were amongst the first plant structures subjected to microscopic study by Robert Hooke (1665) [2]. Reflecting a common sentiment amongst scientists, a contemporary botanist (Prof. W. Barthlott, Bonn University) complained once that research on stinging nettles is a waste of time because generations of botanists have already studied every detail. The morphology and anatomy of stinging hairs has indeed been studied widely, and there is a solid body of knowledge on their structure [1,3]. Nevertheless, we are far from having a comprehensive understanding of stinging hair morphology, function, and toxicological mechanisms, as we intend to show below. Similarly, there has been a wide range of chemical and physiological studies, but their results are fragmentary and contradictory. Overall, stinging hairs—prominent and impressive plant structures—are still not well understood.

The most widely known (and most widely distributed) group of stinging plants are the stinging nettles (genus *Urtica*, Urticaceae). When asked about the active principle in the stings of nettles, any natural scientist will likely almost immediately confirm that (a) formic acid is the active principle and (b) this has been known for years. The assumption that formic acid is the active principle is indeed one of the oldest answers to this question (compare Table 1, from Thurston & Lerston (1969)) [1], which does not make it any more correct.

For the past ca. 150 years, scientists have tried to answer the question of what makes nettles sting at a chemical level, and there are hardly any two studies which reach the same conclusion. Part of the problem is, of course, the very small volume of liquid in the stinging hairs—typically less than 10 nL in *U. dioica* (Table 2)—rendering direct analysis very difficult.

For the detection of neurotransmitters, which were known as pain-inducing substances, highly sensitive methods became available in the mid-20th century. Thus, the substances acetylcholine, histamine, and serotonin could be detected and quantified in extracts of individual stinging hairs by physiological tests on smooth muscle tissue [21,22]. These neurotransmitters are still accepted as active substances in stinging hairs, although they seem not to be sufficient to explain all effects of the nettle stings [27].

Several other authors, using sophisticated analytical techniques such as high-performance liquid chromatography (HPLC) or gas chromatography (GC) in combination with mass spectrometry, based their findings on extracts of entire plant leaves [28]. These provided no clear indication as to which of the compounds identified were even found in the actual stinging trichomes, let alone causative agents for the stinging properties. Moreover, widely different plant species with stinging hairs—especially within the family Urticaceae—were analyzed. Conflicting results are as likely to be the result of different methodologies as of the different plants actually containing different compounds. Thus, it has long been known that the active principle of the sting of the wasp and the sting of the bee shares some common features, but differs in some details: phospholipases, hyaluronidase, and the neurotransmitters serotonin, histamine, dopamine, noradrenaline, and adrenaline are found in both bee and wasp venom, but the toxins differ in their peptide composition, with some peptides characteristic of bee venom and others of wasp venom [29]. Here, we compile our current knowledge about plant stinging hairs and their toxins, trying to highlight what is known and what remains unknown, following up on a relatively recent review by Fu et al. (2007) [28]. We shall discuss the definition and occurrence of stinging hairs in the plant kingdom, their putative active principles, and the ecological roles of plant stinging hairs. We provide some original data and images (and a corresponding Materials and Methods section) for up-to-date illustrations and to put published work in the context of our own ongoing research.

## 2. Stinging Hairs

Plant stinging hairs are fascinating from a range of different perspectives. We will approach the topic by looking first into their morphology and function, then delimiting them from other plant trichomes; by examining their distribution in the plant kingdom and on individual plants; and by reviewing known physiological effects and summarizing our current state of knowledge on their chemical properties.

### 2.1. What Are “Stinging Hairs”?

Plant hairs or trichomes seem to defy classification—there is a nearly infinite number of different shapes, structures, and chemical compositions, rendering any attempt at providing a unified classification futile. Thus, the term “stinging hair” has been variously used in a quite broad definition, including trichomes that, due to their shape and brittleness, break off and cause mechanical damage only. For the purposes of the present review, we follow the much narrower definition of Hewson (2019) [30], which is widely used in pertinent literature such as Thurston & Lerston (1969) [1], and more recently by Fu et al. (2007) [28] and Mustafa et al. (2018) [3], but dividing the stinging hairs into two different types—the *Urtica* type and the *Tragia* type. Stinging hairs of the *Urtica* type are unicellular hairs containing a toxic fluid and associated with a group of specialized basal cells, usually forming a pedestal elevated above neighboring epidermis cells. Aside from variations in the shape of the tip, mature stinging hairs of the *Urtica* type are remarkably uniform and found across Urticaceae, Loasaceae, *Wigandia* (Namaceae), *Horovitzia cnidoscoloides* (Caricaceae), and *Cnidoscolus* (Euphorbiaceae) [1,3]. They do, however, differ dramatically in size, ranging from roughly 1 to 7 mm in length (in the illustrated examples; Figure 1).

Stinging hairs act as hypodermic syringes, injecting their fluid content into the skin with the help of a sharp tip formed upon breakage of the apex. These stinging hairs, in the narrow sense, i.e., injecting irritant fluid, still come in two profoundly different varieties, here termed *Urtica*-type and *Tragia*-type, respectively.

*Urtica*-type stinging hairs.—The type described by Robert Hooke (1665) [2] from *U. dioica* is the most common form and will here be referred to as the *Urtica*-type stinging hair. The basic structure of typical stinging hairs has been investigated in multiple publications since [1,3,7,31], so we will only provide a basic outline here.

The central part of the stinging hair is the stinging cell itself—a very large, living, single cell filled with an irritant fluid. The exposed walls of this cell are thick, mineralized, and stiff, while the cell walls immersed into the pluricellar pedestal are thin and flexible (Figure 2a–e). The mineralized tip—usually bent or globose—of the stinging hair easily breaks off upon touch, e.g., by an animal, leaving an open cannula with sharp rims that can easily penetrate skin and inject the fluid (Figure 2g–k). Pressure upon the stinging hair compresses the flexible bladder-like base of the stinging cell, reducing the internal volume and pressing the fluid out of the open tip—and possibly into the skin of a victim. The function is essentially that of a bladder pipette (Figure 2f). SEM images of cryo-preserved versus air-dried stinging hairs illustrate the stability and rigidity of the mineralized walls, contrasting the flexibility—visualized by shrinkage—of the flexible basal part and pedestal.

Major differences between *Urtica*-type stinging hairs are found in the cell wall material of the exposed part of the stinging hair. The stinging hairs of *Cnidoscolus* differ strikingly from those of all others of the *Urtica* type in that they are entirely unmineralized [3]. All other *Urtica*-type stinging hairs studied could be shown to be variously mineralized, typically with calcium carbonate, calcium phosphate, and/or silica alone, or in different combinations [3]. Due to the widely different sizes of stinging hairs, their internal volume, i.e., the amount of stinging fluid inside the stinging hairs, also varies widely, ranging from roughly 4 nL in *U. dioica* to well over 100 nL in *Cnidoscolus aconitifolius*.

Strikingly, biomineralization of *Urtica*-type stinging hairs is often remarkably divergent between closely related species of the same genus, but relatively similar between widely different species [3]. Conversely, functionality of *Urtica*-type stinging hairs is quite uniform—with the tip breaking off upon contact and the irritant liquid expelled by the compression of the inner volume of the hair—and quite independent of the divergent cell wall chemistry.

“*Tragia*-type” stinging hairs.—Stinging hairs of some Euphorbiaceae are strikingly different: Knoll (1905) [13] and Thurston (1976) [32] described the structure of the stinging emergence of *Tragia* (Plukenetieae) and *Dalechampia* (Dalechampieae). These are also structures made up of several cells, but with the actual stinging cell provided with an apically positioned, sharp-pointed, calcium oxalate crystal in the cell lumen (Figure 3a–c). Upon contact, the distal cell wall breaks and the crystal is ejected forcibly, together with the irritant fluid contained in the cell. This “stinging cell” emerges from a group of elongated jacket-cells. This unique type of “ballistic” stinging hair has so far only been reported from the two closely allied groups of Dalechampieae and Plukenetieae in Euphorbiaceae [33]. These are also clearly stinging hairs as defined above—acting to inject irritant fluid into the skin—but are both morphologically and functionally divergent from the other stinging hairs here considered. Our own recent analyses could confirm the composition of the stinging hair apex as calcium oxalate by Raman spectroscopy and energy dispersive X-ray (EDX) element analyses with an SEM (Figure 3); other parts of the hair and epidermis cells were found to be unmineralized.

Irritant hairs, which do not inject a toxic fluid, may act mechanically and cause some pain when the brittle, often mineralized tips break off and remain in the skin. They are found in, e.g., Boraginaceae such as borage (*Borago officinalis*) and *Symphytum,* Urticaceae (*Forsskaolea*) [34], Solanaceae (*Solanum carolinense*) [35], or as multicellular hairs in Cucurbitaceae (*Cucurbita*) (Figure 4). The term “stinging hairs” has further been used for trichomes that are additionally associated with toxins on the plant surface, causing a reaction on the skin upon contact, e.g., in the velvet bean (*Mucuna pruriens*, Fabaceae, Rosales) [36] or the Barbados cherry and allied species (Malphigiaceae, Malphigiales) [37]. We prefer to differentiate these trichomes for purely mechanical irritation as “irritant hairs” from stinging hairs proper.

### 2.2. Distribution of Stinging Hairs in the Plant Kingdom

The *Urtica*-type stinging hair is the more widespread type. The most well-known group of plants with stinging hairs are the true “nettles” of the family Urticaceae (order Rosales) (Figure 5a,b). In this family, stinging hairs are found in representatives of one subgroup only, the tribe Urticeae. Urticeae includes ca. 10 genera and ca. 150—virtually exclusively stinging—species [34,38]. The stinging nettle proper—*Urtica*—includes the “common nettle” (*U. dioica*) as the most well-known representative, but is a species-rich and widespread genus. It comprises ca. 60 species [39] and is found on all continents and most islands, from subarctic and subantarctic environments, over temperate and Mediterranean zones, to all major tropical mountain ranges, including the East African mountains, the Andes, the Central American Cordillera, and even New Guinea, New Zealand, and Tasmania. Other important Urticeae genera with stinging hairs include *Laportea* (ca. 21 spp.), mainly from SE Asia, and the two tropical, often tree-like or truly arboreous genera, *Urera* (ca. 35 spp.) and *Dendrocnide* (ca. 37 spp., including rainforest trees up to 40 m tall). *Dendrocnide* comprises some of the most violently stinging species in the entire plant kingdom [40].

The much less well-known rock nettles or Loasaceae (order Cornales) are another family with numerous stinging species [41] (Figure 5c,d). Stinging hairs in Loasaceae are widely reported from the subfamily Loasoideae, comprising ca. 200 mostly herbaceous species nearly exclusively in the Americas. Loasoideae comprise a range of genera, nearly all of which possess stinging hairs, with the largest being *Nasa*, with over 100 species, and *Caiophora*, with an estimated 50 species. Loasoideae are primarily Andean in distribution, but there are also a few representatives of the family in, e.g., Argentina, Brazil, and Central America. While some species, such as the mostly Brazilian genus *Aosa*, are fiercely urticant, others appear to have secondarily lost their functional stinging hairs. Stinging hairs are also reported from representatives of the family outside subfamily Loasoideae, e.g., the genus *Eucnide*, but no micromorphological studies have been published confirming their identity as stinging hairs in the narrow sense here employed, versus just sharp hairs doing mechanical damage.

Euphorbiaceae (order Malphigiales) are another family with many urticant representatives. *Cnidoscolus* (tribe Manihoteae) (Figure 5e) is a large genus with nearly 100 species restricted to subtropical and tropical America [42]. *Cnidoscolus* is universally urticant, with the only exception of a non-stinging form of *C. aconitifolius*, which is an important green vegetable in northern Central America (“chaya”) [43]. The other group of Euphorbiaceae with stinging hairs are only remotely related (tribe Plukenetieae) and are provided with *Tragia*-type stinging hairs. *Tragia* (c. 150 spp.) and some allied genera (subtribe Tragiinae, ca. 195 spp. overall) [44] and some species of closely related *Dalechampia* (subtribe Dalechampiinae, ca. 130 spp. overall) [33] (Figure 5f) are usually provided with stinging hairs. Both groups are widespread in the tropics and subtropics.

The final two plant families with genuine stinging hairs are much smaller and have the typical *Urtica*-type stinging hairs. Namaceae (order Boraginales) have a single genus with genuine stinging hairs, namely *Wigandia*, with ca. five species in Central and South America (Figure 5g,h). These are large-leaved, drought-deciduous shrubs or small trees from seasonally arid environments. Closely allied Hydrophyllaceae have several representatives with hard, hollow, mineralized trichomes in the genus *Phacelia* (e.g., *Phacelia malvifolia*), but these appear to be just stiff spiny hairs without the ability to inject caustic liquids. Finally, there is a single species with stinging hairs in the family Caricaceae—*Carica cnidoscoloides* (=*H. cnidoscoloides,* order Brassicales)—from Southern Mexico [45] (Figure 5i).

Overall, there are ca. 650 species of plants with stinging hairs, of which roughly 450 species possess *Urtica*-type stinging hairs and over 200 species feature *Tragia*-type stinging hairs.

### 2.3. Distribution of Stinging Hairs on Plants

Stinging hairs are often found on all or most aerial parts of the plant, including the stem, the petioles and leaves, and the calyx (Figure 6a–c). The corolla and the internal flower parts usually lack stinging hairs, but in Loasaceae (e.g., *Caiophora*), the outsides of petals may be densely covered with stinging hairs, and there may be abundant stinging hairs in the inside of the flower and even inside the ovary (Figure 6d–f). Animal-dispersed fruits that are designed for endochorous dispersals, i.e., for consumption by animals, usually lack stinging hairs on their surface. The fleshy tepals covering the fruits of *Urera* and *Dendrocnide* (Urticaceae) are free of stinging hairs, but the inflorescence axes may be densely covered with stinging hairs. The dry fruits of *Urtica* tend to lack stinging hairs, but some species or varieties have stinging hairs on the tepals enclosing the achenes (Figure 6j). Conversely, the capsular fruits of Loasaceae and *Wigandia* and the explosive capsules of *Cnidoscolus* (active ejection of seeds), and generally the fruits of stinging species of Euphorbiaceae, are always particularly densely setose, evidently to discourage granivores and frugivores (Figure 6g–i).

### 2.4. Homology and Ontogeny of Plant Stinging Hairs

Stinging hairs are remarkably complex morphological features and their origin can best be understood by a comparison to non-stinging hairs and a study of their ontogeny. In most stinging-hair-bearing species, particularly Urticaceae, Loasaceae, and *Cnidoscolus* species, the stinging hairs differ considerably from non-stinging hairs, but may be connected to them by morphological intermediates. Unlike the morphologically homogeneous stinging hairs, the much smaller non-stinging hairs are often highly variable in size and morphology within and between species (Figure 7a–c) [31].

*Urtica*-type stinging hairs are initiated early in organ development, and young organs are, therefore, often densely covered by stinging hairs, becoming more widely spaced after organ expansion. Conversely, non-stinging hairs may develop throughout organ development, keeping up with organ expansion and ensuring a dense and even trichome cover of the plant organ independent of age. Genuine stinging hairs are essentially hollow, including the apical bulb, and are provided with a more or less clear preformed breakage point. In contrast, the small non-stinging hairs of many species are hollow only early in ontogeny; later, the lumen becomes filled with mineral precipitations such as amorphous calcium carbonate [46]. Such massive small trichomes are also common on many other plant groups, e.g., Boraginaceae and Brassicaceae.

Intermediate trichome forms are widely found in, e.g., *Wigandia* (Namaceae) (Figure 7d). Their stinging hairs are morphologically very similar to non-stinging unicellular trichomes on the same plant organ. The latter can reach the size of small stinging trichomes and are provided with only a smaller pedestal of a few foot cells. The genuine stinging hairs are, however, clearly differentiated by the presence of the bulbous tip and the preformed breakage point. Stiff but non-stinging trichomes, such as those of *Nama rothrockii* (Namaceae) and *P. malvifolia* (Figure 7e,f) from closely allied Hydrophyllaceae, also represent intermediates between the two hair types: they are similar in size and mineralization to stinging hairs of *Wigandia*, but they lack the preformed breaking tip and—presumably—any irritant liquid inside.

It can be assumed that *Urtica*-type stinging hairs originated from single-celled, simple trichomes. They seem to be typical unicellular epidermal trichomes in origin, as subepidermal layers of the leaf are only evolved in the formation of the pluricellular pedestal. In some species, such as *Urtica ferox* from New Zealand, this pedestal can actually make up most of the length of the stinging emergence, but the actual stinging hair is still only the unicellular tip (Figure 1d).

The derivation of *Urtica*-type stinging hairs is in stark contrast to the situation in the *Tragia*-type stinging hairs. A detailed study of the ontogeny of the stinging hairs in *Tragia* shows that the actual stinging cell is of subepidermal origin and goes back to typical crystal idioblasts [32]. Crystal idioblasts are individual cells differentiated in a tissue by their usually spherical shape, thin cell walls, and the presence of one or several large crystals. In some plants, specialized crystal idioblasts—so called biforines—are able to forcibly expel the crystal together with the often caustic fluid contained in the cell [47]. This is a clear parallel to the situation in *Tragia* and allied species, where the stinging hair expels both the calcium oxalate crystal and a caustic liquid from the stinging cell. The actual stinging trichome is, thus, a crystal idioblast of subepidermal origin and surrounded by epidermal jacket cells. No comparative study has investigated the distribution of biforines or crystal idioblasts in Euphorbiaceae, and the homology of the specific *Tragia*-type stinging hairs remains enigmatic.

### 2.5. Physiological Effects of Stinging Hairs

The effect of stinging hairs on humans has, of course, been the main point of interest from a scientific perspective. Our own experiences with a range of different species from a range of different families in cultivation show that the reaction to being stung varies widely in type and intensity, ranging from a mild, ephemeral itch to intense pain lasting for well over 24 h, and from no visible effect on the skin to extensive oedema. The most commonly known effect of a contact with stinging hairs is immediate, intense pain—the optimal strategy of the plant to avoid being eaten, and herbivores are likely to quickly learn to avoid these plants. Urticaria with a characteristic visible dermal reaction is another common effect of contact with stinging hairs. Contact with *U. dioica* usually causes a red spot due to capillary dilation, flare (arteriolar dilation), and a welt (exudation of fluid into the tissue). This is also known as “triple response of Lewis” or typical urticaria [48,49,50]. Similar skin reactions can be observed upon contact with many urticant species of Loasaceae, especially of the genera *Nasa, Loasa*, and *Caiophora*. The visible effects may disappear after an hour or so, but may also persist for 24 h or more. Thus, Edwards & Remer (1983) [51] and Coile (1999) [52] reported burning sensation, agonizing pain, and welts persisting for several hours after contact with *Urtica chamaedryoides*. Similar reactions are observed upon contact with other stinging plants, with some of them leading to particularly prominent skin reactions, e.g., *Aosa rupestris* (Loasaceae) and *Urtica flabellata* (Urticaceae). In these cases, extensive blistering may occur, with healing taking weeks rather than hours. As would be expected, mucous membranes are more sensitive to the effect of stinging hairs, but few reports are available of humans with such experiences. One case report indicates a very severe reaction after placing a nettle leaf on the tongue, requiring hospitalization [53].

Stinging hairs may also have neurological effects far beyond the immediate pain and itch. Haberlandt (1886) [7] provided a review of the early reports of the effects of stinging plants. Tropical representatives of the genus *Dendrocnide* featured particularly prominently: “Leschenault de la Tour reported 1819 that after touching *Urtica crenulata* (=*Dendrocnide crenulata*) in the Botanical Garden in Calcutta he saw no weals on the skin, but felt heavy pain and paroxysms for over a day, it took 9 days for the pain to disappear.” Haberlandt (1886) [7] cited similar, possibly exaggerated, reports for *Urtica urentissima* (= *Dendrocnide urentissima*) and *Laportea gigas* (= *Dendrocnide moroides,*
Figure 8 a). MacFarlane (1963) [54] gave a detailed characterization of the “stinging properties” of *Dendrocnide* (under the incorrect name *Laportea*). More recently, Marina Hurley (2000) [55] described the effects of contact with *Dendrocnide*: “The reaction can vary from mild irritation to death, not only in humans, but also in dogs and horses. The burning pain is felt almost immediately after contact, then intensifies, reaching a peak after 20–30 min. During this time, the heart rate increases and the lymph glands in the joints begin to swell and throb, causing almost as much pain as the sting. The hairs are so tiny that the skin will often close over them, making them difficult or impossible to remove.” A man who had fallen into a stinging tree “was literally tied to his hospital bed for three weeks because the pain was so bad” [56]. *Dendrocnide* may cause health problems even without direct physical contact, with its fine trichomes apparently becoming airborne, and may cause strong irritation to the respiratory tract (Figure 8a) [23,55].

*Urtica ferox* is another species provided with particularly vicious stinging hairs (Figure 8b–d). An individual stinging hair causes a painful sting comparable to that of a bee, with pain and numbing lasting for well over 24 h (MW, pers. obs.). *Urtica ferox* is, accordingly, the only other stinging plant for which relatively extensive documentation of its toxicity is available. “*Urtica ferox* neuropathy” has been characterized by abdominal pain, paresthesia, weakness, and incoordination, and takes several weeks to subside [28,57]. Poisonings of horses and dogs, including neurological phenomena and lethal poisoning, have been reported by Connor (1977) [58] and Puig et al. (2019) [59]. In 1961, a man reportedly died 5 h after contact with *U. ferox* with neuropathological symptoms. In other cases, a group of hikers suffered loss of coordination [58], a typist became unable to work for 5 days [60], and a hunter developed several neurological symptoms which were recorded in detail [61]. Similar neurological effects are reported for dogs and horses [51,52] after contact with the American “Fireweed” *U. chamaedryoides*. Dogs suffered nausea, vomiting, and other neurological effects. Humans experienced burning, agonizing pain, and welts persisting for several hours.

Although a range of both dermatological and neurological effects have been reported from stinging plants, the link between the human reaction or disease to the content of the stinging hairs is not always clear, and the plant toxin responsible for a certain pathophysiology has not been unequivocally identified. Massive allergic contact dermatitis is reported as caused by a class of substances called “phacelioids” in *Wigandia caracasana* in Mexico and may be difficult to differentiate from urticaria caused by stinging hairs [62]. However, the study of Reynolds et al. (1986) [62] clearly showed that the causative agent is found in the abundant glandular trichomes and that stinging hairs play no role in this particular case (plant parts without stinging hairs were used for the study). Similarly, details of a report on contact dermatitis from *Cnidoscolus angustidens* argue for the effect of a contact allergen on the leaves rather than the effect of stinging hairs [63]. The fact that *Dendrocnide* can still cause a prominent skin reaction in dried herbarium specimens after decades [56] demonstrates that the administration of liquid in the stinging hairs by the specific “hypodermic needle” mechanism plays a minor role in its toxicity, and the active toxin/agent may also be present on the leaf surface, independently of the stinging hairs.

### 2.6. Toxins of Stinging Hairs

The chemical nature of the irritant in stinging hairs has intrigued scientists ever since the 19th century. Historically, the first plants to be investigated were European representatives of *U. dioica*, the “classical” stinging nettle, and the first substance proposed as the active principle was formic acid (Table 1) [4,6,10,12,16]. The presence of relevant amounts of formic acid has long been disproved [7], but is still widely cited (e.g., Kregiel et al. 2018) [64]. Organic acids have since been repeatedly implicated in the pain-inducing effect of the stinging hairs: acetic and formic acid in *Dendrocnide gigantea* (as *L. gigas)* [14], acetic acid in *Loasa* [8], and recently, oxalic acid and tartaric acid in *Urtica thunbergiana* [28]. However, the proposed role of free organic acids as an active principle literally fails the litmus test: none of the stinging hair fluids tested show a sufficient acidic reaction (own observation). For *Urtica*, the only widely studied genus, a whole range of other substance groups have been proposed in the past, such as glucosides [5,19], enzymes [7,17], tannic acid [9], and alkaloids [11,15,20], but none of these reports have been confirmed with modern methods.

Neurotransmitters.—The first conclusive results showing the identification of substances known as neurotransmitters came in the mid-twentieth century, when acetylcholine and histamine [21], and then serotonin [22], were identified in the sap of the stinging hairs of *Urtica*. These neurotransmitters play a crucial role in signal transduction in animals (including humans) and display a wide range of physiological effects, with histamine, for example, playing a crucial role in neuropathic pain. These three substances were subsequently reported from two other urticant taxa of the Urticaceae, namely *D. moroides* (as *L. moroides*) [23] and *Girardinia* [25,26]. These three “neuromediators” are still accepted as major constituents of the stinging hairs of some *Urticaceae*. The presence of histamine has since also been demonstrated in *Cnidoscolus oligandrus* by Cordeior et al. (1983) [65], but Willis (1969) [66] could not confirm its presence in *Cnidoscolus stimulosus*. According to our measurements, histamine and serotonin can also be detected in the stinging hair liquid of Loasaceae by mass spectrometry with nano-electrospray ionization. Spectra of *Loasa heterophylla* and *Caiophora deserticola* indicate strongly varying concentrations of these compounds. The spectrum of *L. heterophylla* shows an intense peak of histamine, but only traces of serotonin. In *C. deserticola*, the most intense peak indicates serotonin, whereas histamine occurs only in traces (Figure 9). Acetylcholine has not yet been found in these Loasaceae samples, nor its possible decomposition products such as betain. These strong variations within Loasaceae are similar to results of preliminary measurements of some Urticaceae [67].

In summary, the presence of one or more of these neurotransmitters in many stinging hairs can be considered as well confirmed, and substances such as histamine certainly provide a partial explanation for the pain and the triggering of contact urticaria itself [50]. Divergent analytical results may go back to the limited stability of these compounds and their possible enzymatic degradation upon tissue damage. Collier and Chesher (1956) [22] reported the decay of serotonin within 30 min, and acetylcholine is very rapidly hydrolyzed by cholinesterases [68]. In the animal body, neurotransmitters have a range of functions associated with nerve cells and are often mediated by actions on ion channels and ion pumps. Many of the neurotransmitters have also been documented from plant tissues, where they also act in regulating ion channels and membrane potentials, although this is obviously not associated with nerve cells [68,69,70,71]. Tretyn & Kendrick (1991) [72] argue: “It is proposed that the primary mechanism of action of acetylcholine in plant cells is via the regulation of membrane permeability to protons (H^+^), potassium ions (K^+^), sodium ions (Na^+^) and Ca^2+^.” The widespread occurrence of neurotransmitters such as acetylcholine and histamine in plants indicates that they did not specifically evolve for plant defense, but rather have been secondarily weaponized in stinging hairs. However, it seems unlikely that these substances alone explain all the divergent effects of the contact with stinging hairs across the different plant species and families.

Three additional groups of candidate substances have since been proposed: leucotrienes, peptides, and proteins. Studies by Czarnetzki et al. (1990) [73] reported the presence of immunoreactive leukotrienes in *Urtica urens*—a result that has not been re-investigated more recently. No additional reports of leukotrienes from *Urtica* or other stinging hairs are known to us, but the concentrations of 0.15 to 0.3 pg per hair reported by Czarnetzki et al. (1990) [73] are apparently much too low to cause the typical reaction. Juhlin & Hammerström (1983) [74] demonstrated that 0.01 to 1 ng is necessary to produce a skin reaction.

Peptides and proteins.—A striking discovery was the identification of the octapeptide Moroidin from the Australian gympie bush (*D. moroides*) by Robertson & MacFarlane (1957) [23]. The molecular structure was confirmed by Kahn et al. (2000) [75], but a direct link between this substance and the effect of *Dendrocnide* stinging hairs could neither be demonstrated by these authors nor any study since. Leung et al. (1986) [40] cast serious doubt on the pain-producing activity of Moroidin.

*Dendrocnide* has since received additional attention as arguably the most potent stinging agent of all stinging plants. Moreover, a recent study managed to unequivocally identify an active principle. The pain-inducing and toxic effects of *D. moroides* are unique among stinging plants in that dry leaves retain their irritant properties for a very long time: “The toxin is stable and heat resistant and retains its pain-producing properties for decades. Dried botanical specimens collected over 100 years ago, can still sting” [56]. This indicates the presence of an unusually stable irritant, ruling out, e.g., the three classical neurotransmitters. Gilding et al. (2020) [76] analyzed the content of stinging hairs of two species (*D. moroides* and *D. excelsa*) by HPLC and MALDI-MS, identifying a single fraction of ultrastable miniproteins of ca. 4 kDA as responsible for the physiological effect. They dubbed the group of structurally similar miniproteins “gympietides” after the gympie-bush (the local name of *Dendrocnide*). Extensive experiments demonstrated that gympietides could comprehensively explain various well-known features of *Dendrocnide* toxicity. Importantly, synthesized gympietides could replicate the effect of those isolated from plants, underscoring that they are at least the primary active principle, if not the only one. The discovery of gympietides is thus the first documentation of a genuine neurotoxin in plant stinging hairs.

Inorganic ions.—The possible presence of inorganic ions in the stinging hair fluid has received little attention in the past, partly because stinging hair liquids are usually diluted in buffer solutions for analysis, precluding the subsequent identification of potassium or phosphate. Our own observations were carried out in the scanning electron microscope with an attached EDX (energy-dispersive analysis with X-rays) system. A single dried droplet of stinging hair fluid can be viewed and is sufficient to be analyzed in this way. Figure 10 and Table 3 show element spectra of dried fluids of several species. They contained potassium (K) salts in varying concentrations (together with organic carbon), often as potassium phosphate in Urticaceae (*U. dioica, U. massaica*), or chloride in *Urera baccifera*. The SEM image of the droplet of *U. baccifera* shows dendritic crystal complexes of KCl in the homogeneous matrix.

These preliminary results indicate the presence of considerable amounts of inorganic potassium salts in the stinging hair fluids. We have not been able to quantify the potassium concentrations (due to the uncertain amount of water and inaccuracy in measurement of the lighter elements C and O), but a physiological effect is conceivable. Armstrong et al. (1951, 1953) [77,78] demonstrated a pain-inducing activity for 0.1% KCl via the “blister tests for cutaneous pain,” and thus potassium itself might act as a pain-inducing substance via hyperosmotic stress. Equally plausibly, there could be a synergistic effect of even low concentrations of potassium ions in the presence of neurotransmitters such as acetylcholine as a regulator of ion channels.

There does not seem to be a single or a simple answer to the question of the active principles of plant stinging hairs. The recent description of gympietides [76] represents a breakthrough and is by far the most comprehensive characterization of a likely toxic agent of *Dendrocnide*, explaining also the exceptional neurotoxicity and long-lasting effects. Neurotransmitters have been reported several times in Urticaceae, Loasaceae, and *Cnidoscolus* (Euphorbiaceae) and our own results indicate a possible synergistic effect of inorganic (potassium) ions. There can also be little doubt that histamine plays a prominent role in the dermatological effects of the contact with many stinging plants. However, studies on Urticaceae outside *Dendrocnide* are highly fragmentary, and reports of neurotoxicity within the genus and lasting pain from contact with species such as *U. dioica*, *U. ferox*, and *U. chamaedryoides* strongly indicate that substances beyond neurotransmitters and inorganic ions are involved [52,57,79].

The chemistry and toxicology of plant stinging hairs is thus only tentatively understood. Only a handful of the 450 species with *Urtica*-type stinging hairs have been critically studied with modern, sensitive techniques. No reliable data at all are available on Namaceae, *Cnidoscolus*, or the ca. 200 species with *Tragia*-type stinging hairs.

### 2.7. Ecology of Stinging Plants

The specific ecological role of stinging hairs has been repeatedly studied, albeit not always with the necessary scientific rigor. The overall evidence emerging points to a primary or exclusive role of stinging hairs as a fairly specific defense against mammalian herbivory. Stinging plants across the families are widely avoided by grazing and browsing animals, principally mammals, with common species such as *U. dioica* and *U. urens* often left standing even in overgrazed situations (see Figure 11a–c). *Urtica chamaedryoides* is considered a troublesome pasture weed in the southern United States [52]. There are also reports of grazing favoring stinging Euphorbiaceae in pastures [80,81]. Consequently, some of the stinging plants are typical pasture weeds. At the same time, stinging hairs do not deter all herbivores, and a wide range of mammals include them in their diets in smaller or larger quantities. Species of *Urtica* are reportedly consumed by rabbit (*Oryctolagus cuniculus*), sheep (*Ovis aries*), red deer (*Cervus elaphus*), nuntjac (*Muntiacus reevesi*), and roebuck (*Capreolus capreolus*) [82,83], European bison (*Bison bonasus*) [84], Asiatic Black Bear (*Ursus thibetanus*) [85], and wild pigs (*Sus scrofa*) in Texas [86]. There is evidence that *Urtica* was part of the diet of a range of extinct moa species (*Dinornis robustus*, *Megalapteryx didinus*, *Pachyornis elephantopus*) in New Zealand [87] and of the extinct wooly mammoth (*Mammuthus primigenius*) in Russia [88]. In tropical Africa, the extremely urticant leaves of *Urera* (especially *Urera hypselodendron*) are consumed in quantity by the mountain gorilla (*Gorilla beringei*) [89,90] and chimpanzees (*Pan troglodytes*) [91]. However, there seems to be a dynamic ontogenetic and evolutionary response to herbivory by stinging plants and this has been repeatedly demonstrated. In *U. dioica*, mechanical damage and grazing lead to the formation of leaves with a higher number of stinging hairs [92], and the same is true for *U. thunbergiana* [93] and *C. aconitifolius* [94]. At a selection level, higher grazing/browsing pressure favors plants with more stinging hairs in the population, leading to population divergence. Considerable differences among populations with high herbivore pressure have been demonstrated for *U. thunbergiana* from areas more or less heavily browsed by Sika deer (*Cervus nippon*) in Japan [95] and for *Cnidoscolus texanus* with different grazing regimes [96]. The overall picture emerging is one of mammalian (rarely avian) herbivores as the driving force of an arms race with stinging plants.

Conversely, stinging hairs appear to play a negligible role in the defense against invertebrate herbivores (Figure 12a–d). *Urtica* is a notoriously important species for invertebrates across the temperate zones of the world. Studies in Central Europe reported a total of 100 species of insects from six orders, including 31 more or less obligate phytophagous insects, on *U. dioica* [97]. It has also been demonstrated that *U. dioica* is highly palatable to slugs [98], and an entire subcosmopolitan group of butterflies (Nymphalini) comprises numerous taxa more or less specialized on *Urtica* as larval food plant [99]. Fourteen species of insects were reported to feed on the leaves of *Wigandia* in a single study [100]; *Cnidoscolus* is attacked by specialized sphingid larvae removing its stinging hairs with surgical precision and then interrupting the flow of the sticky, poisonous latex mechanically before consuming the leaf [101] (Figure 12a,b). Even *Dendrocnide*—arguably the most vicious stinging plant—has its own specialized herbivore in a species of beetle (*Diphycephala pygmaea*, Scarabaeidae) [102]. There are few published observations on herbivory in stinging Loasaceae. However, our own observations in the field and cultivation underscore that mammalian herbivores leave them untouched even in heavily grazed habitats, yet they are subject to a large number of invertebrate pests; they are the preferred larval food plants for presumably specialized Pyralidae butterflies; and they are heavily attacked by slugs and snails. In many stinging plants, the stinging hairs are accompanied by additional mechanical defenses in the form of stiff, usually mineralized trichomes. These may be simple and unbranched (*Urtica*, Figure 7a) or morphologically complex and highly varied (*Loasa*, Figure 7c). Additional external chemical defenses appear to be present in most stinging plants and are presented on glandular trichomes, which are nearly always found in the company of “simple” and stinging hairs on the plant surface. The nature of the chemical defensive substances is usually unknown but ranges from oily secretions likely physically “gumming up” the mouthparts of insects in *Nasa* (Figure 12e) to complex defense molecules such as the phacelioids reported from glandular trichomes in Namaceae (Figure 4i) [62]. Beyond external chemical and mechanical defense mechanisms, there is a plethora of internal mechanisms likely playing a crucial role in the defense against invertebrate and/or vertebrate herbivores, such as the complex iridoid compounds in Loasaceae [103] and the caustic latex and extrafloral nectaries in Euphorbiaceae [101,104]. Stinging hairs alone are thus just one weapon in a whole arsenal, apparently with mammalian herbivores as the primary target.

## 3. Conclusions

Plant stinging hairs, especially the *Urtica*-type stinging hairs, are a striking example of convergent evolution and have arisen several times in very distantly related plant groups independently. Stinging hairs are found in ca. 650 flowering plant species across five different plant families belonging to distantly related orders. In spite of their morphological similarity, stinging hairs show striking differences in detail, even beyond the gross distinction of *Urtica*-type and *Tragia*-type stinging hairs, with different degrees and types of cell wall mineralization, massive size differences, and—presumably—divergent chemical compositions. Several aspects of plant stinging hairs can be considered as well-understood, first and foremost their morphology and anatomy, which have been studied repeatedly in the past ca. 350 years. *Urtica*-type stinging hairs of Loasaceae have been shown to have been derived from simple, mineralized, unicellular trichomes [3], and a similar evolutionary trajectory can be assumed for *Urtica*-type stinging hairs in the other plant families. In many species, especially within Urticaceae and Loasaceae, morphological intermediates are found, underscoring this point. Conversely, *Tragia*-type stinging hairs appear to have an altogether different evolutionary origin and go back to crystal idioblasts or, more specifically, endoforins.

The ecological role of plant stinging hairs across the different plant families and across their geographical range seems to be predominantly in herbivore defense and primarily aimed at mammalian herbivores, which makes them successful pasture weeds. This defense, however, is not absolute, i.e., discouraging herbivores but not eliminating herbivory altogether. There is clear evidence that mechanical damage, such as inflicted by browsing, increases stinging hair density on new growth and that populations under higher herbivore pressure evolve higher stinging hair densities, underscoring the direct relationship between stinging hairs and mammalian browsing. At the same time, a very wide range of mammals—including rodents, ursids, cervids, bovids, and primates—consume stinging plants, especially of the nettle family (Urticaceae), often to a considerable degree, demonstrating the limits of this defensive strategy. Stinging hairs, like all plant defenses, are just one weapon in a larger arsenal. They are therefore universally complemented by additional chemical and/or mechanical defenses such as simple, mineralized, or non-mineralized trichomes, and also with secretory, glandular trichomes or poisonous secondary compounds in the plant tissue.

We thus have a fairly good understanding of the distribution, evolution, and ecological role of plant stinging hairs. This is in stark contrast to our highly rudimentary understanding of their actual chemical composition and their toxicology. The physiological effects of stinging hairs—primarily on humans—have been widely described in a variety of publications across the last ca. 150 years. The effects vary dramatically, from a slight skin itch to a massive neurological disorder to—in very rare cases—death. It seems highly unlikely that one and the same substance or set of substances is present across all stinging plants, both in view of the fact that stinging hairs are found in ca. 650 species belonging to five distantly related flowering plant families and that the physiological effects are so divergent. It is thus not surprising that the various studies published over the past 200 years came to widely different conclusions with respect to the active principle when investigating different species from different genera and families.

After reviewing the pertinent literature, most substance classes that have been proposed in the past can be ruled out, and even the proposed acidic nature of the fluid and free formic acid content cannot be confirmed. It seems justified to conclude that neurotransmitters such as histamine, acetylcholine, and serotonin are active compounds at least in many Urticaceae and Loasaceae. Our own—highly preliminary—data indicate that elevated concentrations of inorganic ions such as potassium are present and may have a synergistic effect. Neurotransmitters explain some, but not all, of the effects of stinging hairs on humans [28,48]. They do not explain the massive neurological effects of some representatives of Urticaceae—especially *Dendrocnide*, but also other taxa such as *U. ferox* and some Loasaceae. The recent discovery of the substance class of gympietides in *Dendrocnide* as the first genuine neurotoxin in plants [76] represents an exciting new development and the first real progress since the discovery of the neurotransmitters in stinging hairs ca. 70 years ago. We may thus be beginning to understand the chemistry of at least Urticaceae stinging hairs.

## 4. Outlook

Many aspects of stinging hairs in plants are well understood (especially morphological characteristics), but the details of their chemistry and toxicity are clearly still poorly documented. Based on the recent discovery and characterization of gympietides, a broader screening for these and similar compounds across the 10 genera and ca. 150 species of stinging Urticaceae would be an obvious next step. Additionally, a detailed study of overall composition of stinging fluids and the possible additional presence and role of neurotransmitters and inorganic ions would be highly desirable. One of the very important missing steps is a precise quantification of the individual compounds to improve our understanding of their activities and possible synergistic effects. While we may thus finally have a handle on Urticaceae toxicity, the same is not true for the remaining ca. 500 stinging plant species across four plant families (Namaceae, Euphorbiaceae, Loasaceae, or Caricaceae) in as many plant orders. Especially, Loasaceae often show strikingly similar physiological effects to those found in Urticaceae, including lasting pain and skin irritation.

Technical challenges in the analysis of stinging hair fluids abound. The tiny volume of only a few nanoliters per stinging hair, and therefore the likely minute total amount of active compound, are the most important challenges. Every individual analytical technique will introduce some kind of bias, depending on the size, polarity, solubility, and stability of the compounds investigated [105]. Fortunately, up-to-date mass-spectrometry techniques are sensitive enough to analyze the amounts of liquid found in individual trichomes, permitting the identification of the sum formula not only of small organic molecules. Electro-spray ionization (ESI) is a very gentle method to generate ions for the mass spectrometer, with minimal fragmentation, so that spectra can be acquired from a few nanoliters of complex samples without pre-separation of the components [106]. ESI-MS can detect molecules over a wide mass range, from <100 up to >100,000 Dalton, and is useful even for detection of peptides and proteins. Moreover, ESI-MS can be combined with HPLC and molecule fragmentation techniques for the identification of single components. These technical advances will have to harnessed to come to a better understanding of plant stinging hairs and their physiological effects.

## 5. Materials and Methods

Plants used for our microscopic examination and analyses were grown in the Botanical gardens, University of Bonn, Germany, except for *Horovitzia* and *Dalechampia*, which were kindly provided by Botanischer Garten München-Nymphenburg, Germany, and *Dendrocnide*, which was kindly provided by Hortus Botanicus, Amsterdam, The Netherlands. Scanning electron microscopy was performed with a Cambridge Stereoscan 200 (Cambridge Instruments Ltd., Cambridge, UK), and a LEO 1450 SEM (Zeiss, Oberkochen, Germany), equipped with a custom-made cryo-stage and an Oxford EDX system (Oxford Instruments, Abingdon, UK) for element analyses. Details of sample preparation, microscopy, analyses, and image processing were recently published elsewhere [107,108]. Raman spectra were collected with a confocal Horiba Scientific LabRam HR800 Raman spectrometer (Edison, NJ, USA) at the Institute of Geosciences of the University of Bonn, Germany. Mass spectra were measured with an Obitrap XL mass spectrometer (Thermo Fisher Scientific, Waltham, MA, USA) at the Kekule Institute of Organic Chemistry and Biochemistry, University of Bonn. Fluid from dozens (20 to 100) of stinging hairs was diluted in 10 µL of dd water and stored at −80 °C. Another 10 µL of acetonitrile (HPLC-grade, VWR Chemicals, Darmstadt, Germany) was added immediately before the measurement. Substances were identified via accurate mass and higher energy collision-induced dissociation (HCD) of mass-selected ions.

## Figures and Tables

**Figure 1 toxins-13-00141-f001:**
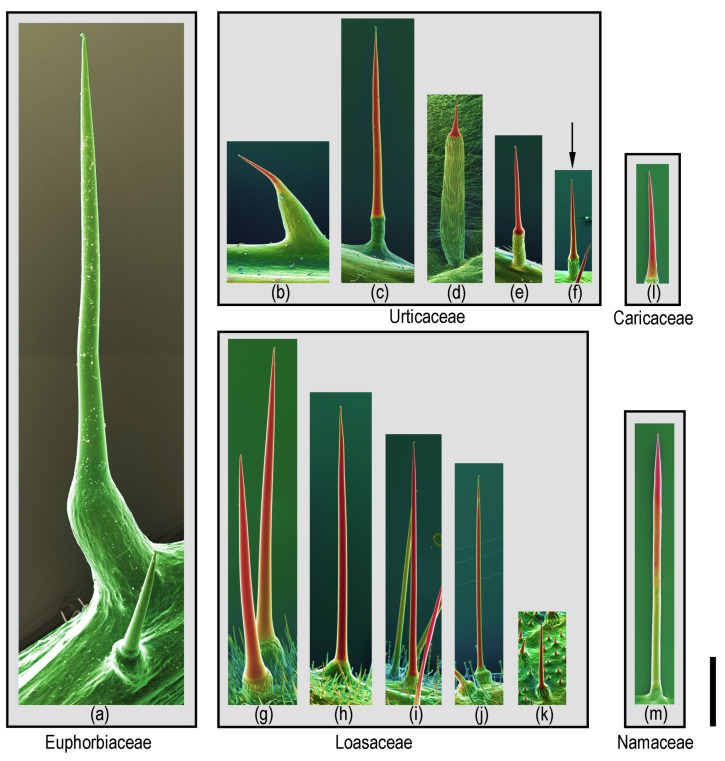
Size comparison of different *Urtica* type stinging hairs: Compositional-contrast scanning electron microscope (SEM) images of fresh or frozen hydrated (cryo-SEM) stinging hairs from different plant families. Euphorbiaceae: (**a**) *Cnidoscolus aconitifolius*. Urticaceae: (**b**) *Urera baccifera*, (**c**) *Urtica mairei*, (**d**) *Urtica ferox*, (**e**) *Urtica atrovirens*, and (**f**) *Urtica dioica* (arrow; one of the smallest ones). Loasaceae: (**g**) *Nasa amaluzensis*, (**h**) *Caiophora deserticola*, (**i**) *Loasa insons*, (**j**) *Aosa rupestris*, and (**k**) *Chichicaste grandis*. Caricaceae: (**l**) *Horovitzia cnidoscoloides*. Namaceae: (**m**) *Wigandia ecuadorensis*. Red color indicates mineralization with high concentrations of of Si or Ca. Scale bar = 1 mm.

**Figure 2 toxins-13-00141-f002:**
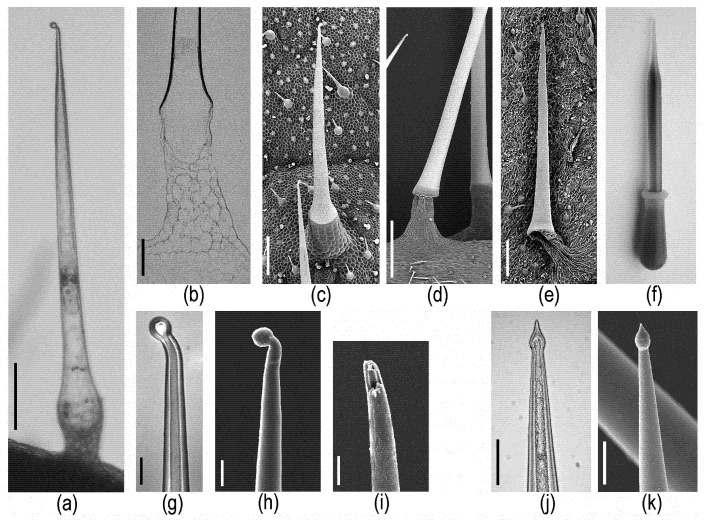
Morphological details of stinging hairs of the *Urtica* type (including *Wigandia*). (**a**–**i**) *Urtica* stinging hairs. Transmission light microscopy (LM) image of (**a**) entire hair and (**b**) section of basal part (SEM block face image) reveal the extent of the single stinging cell. SEM images of (**c**) critical-point- (CP-) dried and (**d**,**e**) air-dried stinging hairs demonstrate the rigidity of the strongly mineralized shaft in contrast to the flexibility of the shrinking non-mineralized tissue of the pedestal. (**f**) The function of a pipette imitates the ejection of fluid from a stinging hair. (**g**) LM image of the tip; the fluid-filled lumen extends into the bulb. (**h**,**i**) SEM images of intact and broken tip. (**j**,**k**) LM and SEM images of *Wigandia* stinging hair tips with slightly different shape, but also hollow up to the bulb. Scale bars: (**a**,**c**,**e**) 200 µm; (**b**) 100 µm; (**d**) 500 µm; (**g**,**h**,**i**) 20 µm; (**j**,**k**) 50 µm.

**Figure 3 toxins-13-00141-f003:**
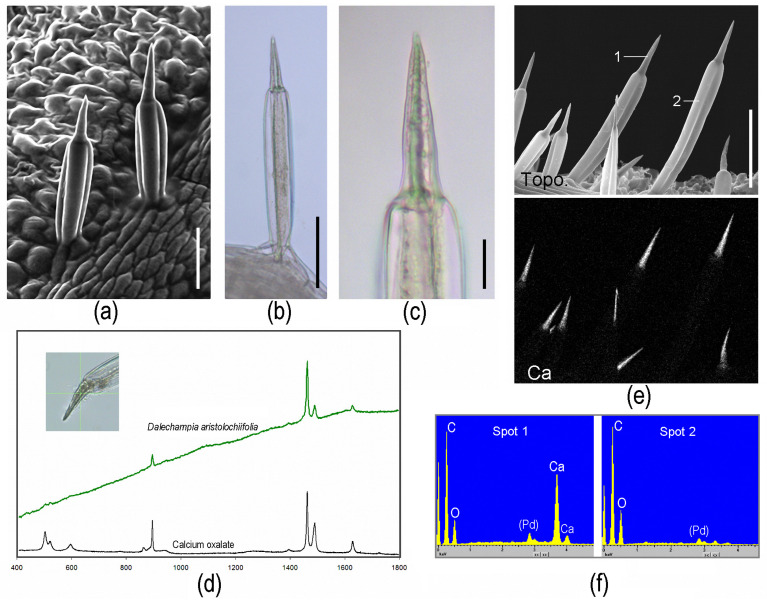
Morphological and chemical details of *Tragia*-type stinging hairs on *Dalechampia aristolochiifolia* plants. (**a**) Cryo-SEM image. (**b**,**c**) LM images, showing an optically dense (or intransparent) structure in the center of the apical part. (**d**) Raman spectrum of the dense structure in the tip; the peaks match perfectly with those of calciumoxalate-monohydrate. (**e**) Topographic image and calcium distribution image by EDX element mapping, showing high Ca concentration in apex. (**f**) EDX spectra of the locations marked in (**e**). (Image (**a**) Yaron Malkowsky, Nees Institut). Scale bars: (**a**,**b**) 50 µm; (**c**) 10 µm; (**e**) 100 µm.

**Figure 4 toxins-13-00141-f004:**
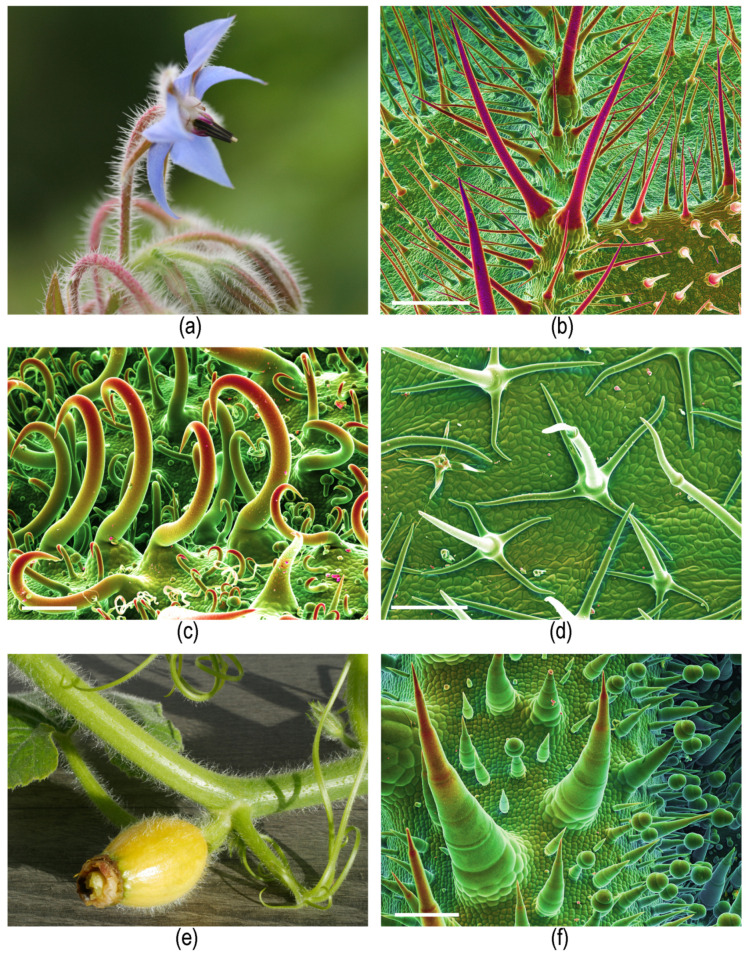
Irritant hairs. (**a**,**b**) *Borago officinalis* (Boraginaceae) showing (**a**) plant habit (flower) and (**b**) SEM image of leaf underside covered with pointed mineralized hairs. (**c**) *Forskaolea angustifolia* (Urticaceae) with mineralized, hook-shaped irritant trichomes. (**d**) *Solanum carolinense* (Solanaceae) with small, unmineralized, branched irritant trichomes; (**e**,**f**) *Cucurbita pepo* (Cucurbitaceae) showing (**e**) plant habit and (**f**) SEM image of multicellular mineralized hairs on leaf underside. (**b**,**c**,**d**,**f**) Compositional contrast cryo-SEM image. Own images except (**d**) Yaron Malkowsky, Nees Institute). Scale bars: (**b**) 500 µm; (**c**) 100 µm; (**d**,**f**) 200 µm.

**Figure 5 toxins-13-00141-f005:**
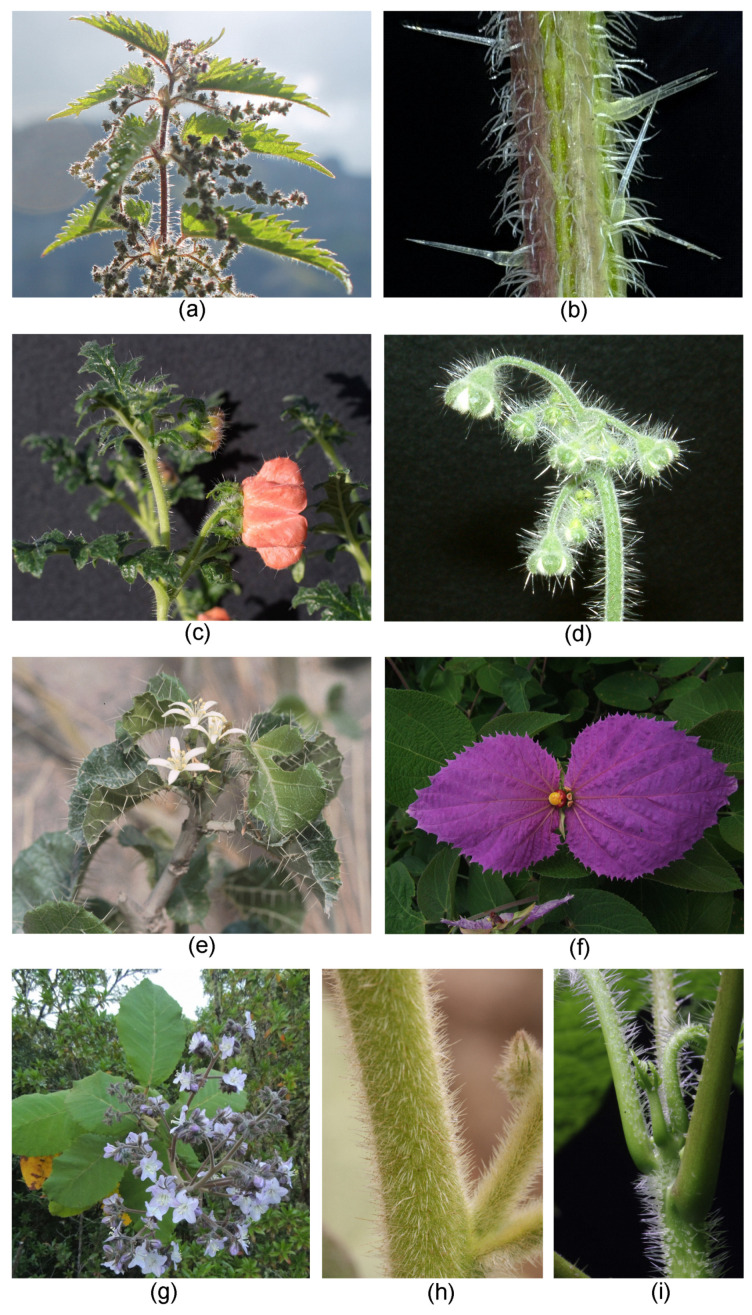
Plant groups with stinging hairs. (**a**,**b**) *Urtica dioica* subsp. *dioica* (Urticaceae) female shoot of the particularly urticant “var. *hispida*” from the Swiss alps. (**c**) *Caiophora deserticola* (Loasaceae) flowering shoot. (**d**) *Aosa rupestris* (Loasaceae) young inflorescence. (**e**) *Cnidoscolus liesneri* (Euphorbiaceae). (**f**) *Dalechamica aristolochiifolia* (Euphorbiaceae) inflorescence. (**g**,**h**) *Wigandia caracasana* (Namaceae) showing (**g**) flowering shoot and (**h**) stem with dense cover of stinging hairs. (**i**) *Horovitzia cnidoscoloides* (Caricaceae) shoot apex. Own images, apart from (**a**) Bernadette Grosse-Veldmann, Nees Institut; (**c**) Markus Ackermann, Koblenz University; (**f**) Günther Gerlach, Botanischer Garten München-Nymphenburg; (**g**) Rafael Acuña, San José, CR; (**h**) Hartmut Hilger, Berlin.

**Figure 6 toxins-13-00141-f006:**
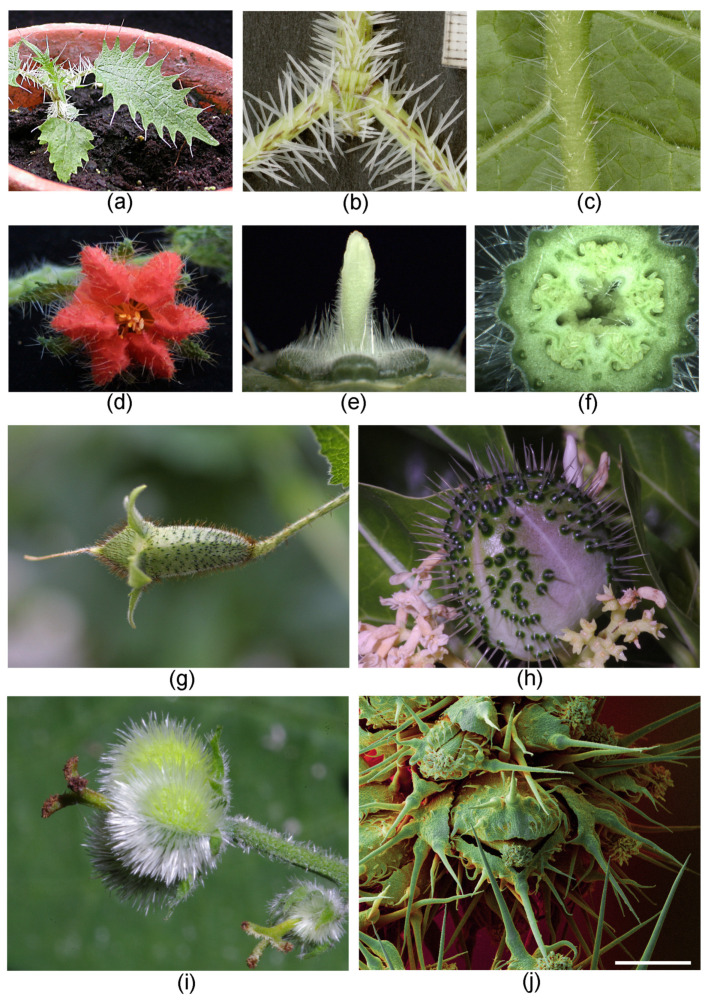
Stinging hair distribution on plants. (**a**) *Urtica ferox* (Urticaceae), young plant. (**b**) *Urtica ferox* (Urticaceae) stem and petioles. (**c**) *Aosa rupestris* (Loasaceae) principal vein abaxial. (**d**) *Caiophora andina* (Loasaceae) flower with dense stinging hair cover on petals. (**e**) *C. deserticola* stinging hairs on receptacle. (**f**) *C. chuquitensis* stinging hairs in ovary receptacle. (**g**) *Nasa macrothyrsa* (Loasaceae) with dense cover of stinging hairs (dark brown) on ovary. (**h**) *Cnidoscolus quercifolius* (Euphorbiaceae) young fruit with extremely dense cover of stinging hairs. (**i**) *Tragia* sp. (Euphorbiaceae) young fruit with extremely dense cover of stinging hairs. (**j**) *Urtica dioica* (Urticaceae) tepals on fruit with stinging hairs. Own images apart from (**a**,**b**) Nicolai M. Nürk, Heidelberg; (**d**–**g**) Markus Ackermann, Koblenz; (**h**,**i**) Günther Gerlach, München. (**j**) False-color SEM image; scale bar = 500 µm.

**Figure 7 toxins-13-00141-f007:**
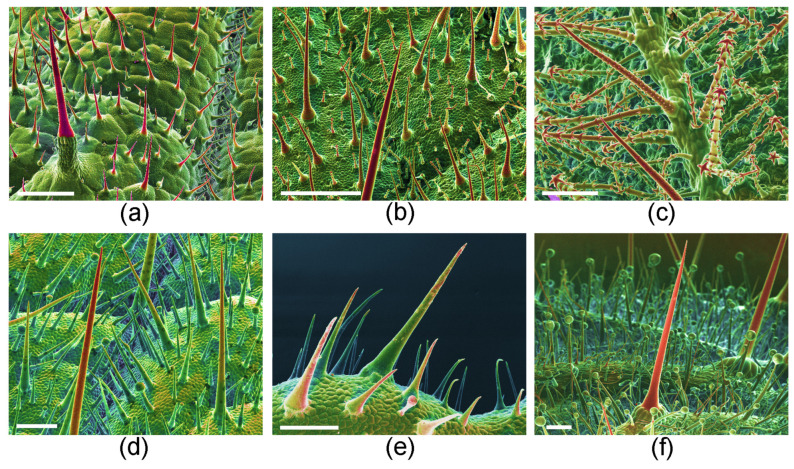
Homology of stinging hairs with other plant hairs. (**a**) *Urtica dioica* (Urticaceae) adaxial leaf surface with stinging hair (red, long pedestal, front left), shorter mineralized, unicellular trichomes (red, small pedestal), and minute glandular trichomes (spheroidal, green). (**b**) *Caiophora lateritia* (Loasaceae) with shorter pointed scabrid hairs and very small glochidiate hairs. (**c**) *Loasa pallida* (Loasaceae) with glochidiate and scabrid hairs on abaxial leaf surface. (**d**) *Wigandia caracasana* (Namaceae) with stinging hairs and shorter, stiff, unicellular trichomes on abaxial leaf surface. (**e**) *Nama rothrockii* (Namaceae) with stiff unicellular trichomes. (**f**) *Phacelia malvifolia* (Namaceae) with stiff, mineralized trichomes (red) and shorter, non-mineralized, glandular trichomes. Cryo-SEM compositional-contrast images, with red portion mineralized. Scale bars: (**a**,**b**) 500 µm; (**c**–**f**) 200 µm.

**Figure 8 toxins-13-00141-f008:**
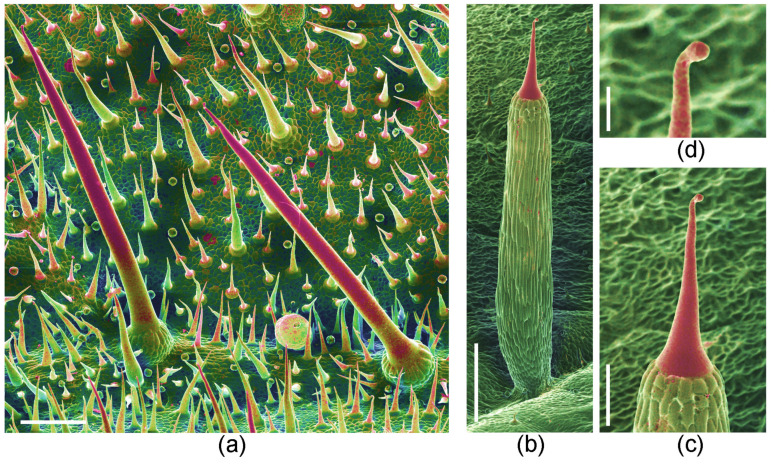
Trichomes of particularly toxic representatives of Urticaceae. (**a**) *Dendrocnide moroides* stinging hairs, shorter stiff trichomes (both mineralized), and minute, spherical glandular trichomes. (**b**–**d**) *Urtica ferox*, with (**b**) very long pedestal crowned with minute stinging hair, (**c**) stinging hair close-up, and (**d**) bulbous tip of stinging hair with pre-formed breakage point. Compositional-contrast SEM images, red portion mineralized. Scale bars: (**a**,**c**) 200 µm; (**b**) 500 µm; (**d**) 50 µm.

**Figure 9 toxins-13-00141-f009:**
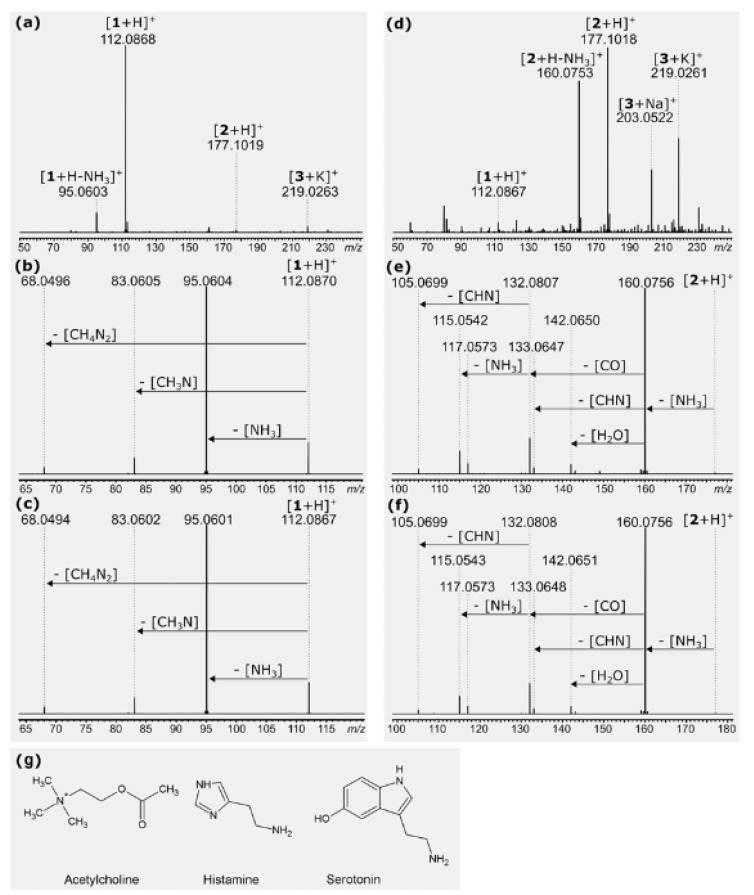
Nano-electrospray ionization mass spectra of stinging hair liquid of *Loasa heterophylla* (**a**–**c**) and *Caiophora deserticola* (**d**–**f**). Assignments: 1 = histamine, 2 = serotonin, 3 = a monosaccharide C_6_H_12_O_6_. (**a**,**d**) Overview spectra, with only the low-mass region shown. Substances are identified via accurate mass and higher energy collision-induced dissociation (HCD) of mass-selected ions. (**b**) Fragmentation of *m/z* 112 from (**a**) spectrum in comparison to (**c**) fragmentation of a reference sample of histamine. (**e**) Fragmentation of *m/z* 177 from (**d**) spectrum in comparison to (**f**) fragmentation of a reference sample of serotonin. (**g**) Structure formulas of the neurotransmitters acetylcholine, histamine, and serotonin.

**Figure 10 toxins-13-00141-f010:**
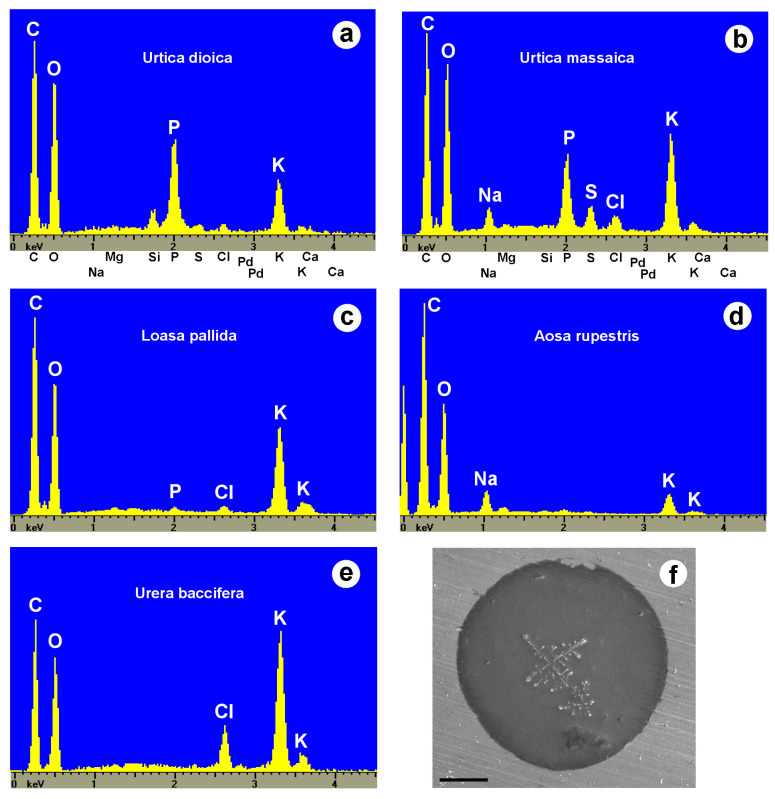
(**a**–**e**) EDX element spectra of stinging hair fluid dry mass from various species. (**f**) SEM image of a dried droplet of *Urera baccifera* stinging hair fluid with dendritic KCl-crystal complexes. Scale bar = 200 µm.

**Figure 11 toxins-13-00141-f011:**
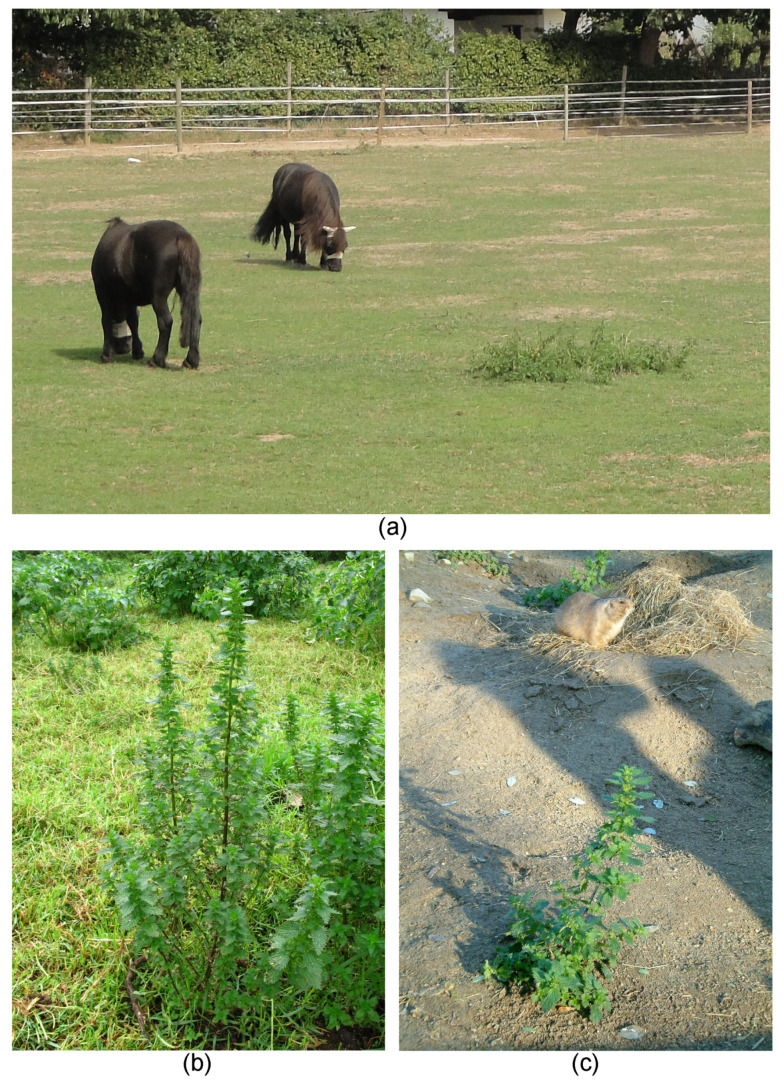
Stinging plants and herbivory. (**a**) Overgrazed pasture with ponies, with only *Urtica dioica* (Urticaceae) left standing (near Bonn, Germany). (**b**) *Urtica urens* (Urticaceae) in a sheep pasture; note closely cropped grass next to untouched nettle (Mooi River, RSA). (**c**) *Urtica urens* as only surviving plant in an enclosure for prairie dogs (*Cynomys*, Tierpark Friedrichsfelde, Berlin. Germany). Own images apart from (**b**) Alexandra Krühn, Berlin.

**Figure 12 toxins-13-00141-f012:**
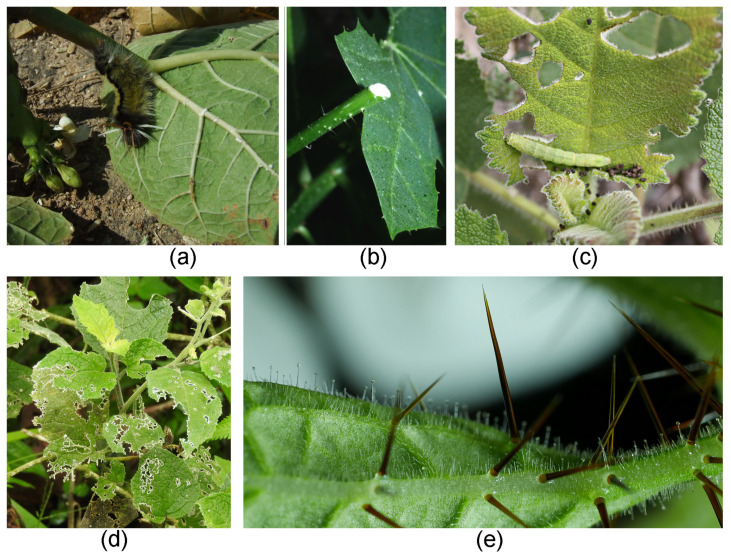
Stinging plants and herbivory. (**a**) Caterpillar feeding on leaves of *Cnidoscolus* (Euphorbiaceae, Northern Peru, Cajamarca); note latex seeping from leaf vein. (**b**) Poisonous latex in *Cnidoscolus* leaf stalk. (**c**) Caterpillar feeding on leaves of *Wigandia* sp. (Namaceae) in Peru. (**d**) Skeletized leaves of *Nasa rubrastra* (Loasaceae) in cloud forest (Ecuador, Tungurahua). (**e**) *Nasa amaluzensis* with brown stinging hairs and glandular hairs producing oily secretions. Own images apart from (**d**) Rafael Acuña, San Jose, Costa Rica.

**Table 1 toxins-13-00141-t001:** Historical reports of chemical constituents in plant stinging hairs.

Authors (Listed Chronologically); Ref.	Formic Acid	Acetylcholine	Histamine	5-hydroxytryptamine	Alkaloid	Acetic Acid	Enzyme	Glucoside	Protein	Tartaric Acid	Resin Acid	Calcium	Salt	Calcium-Sensitive Factor
Hooke (1665); [2]													Urt	
Gorup-Besanez (1849); [4]	Urt													
Rauter (1872); [5]								Urt						
Bergmann (1882); [6]	Urt													
Haberlandt (1886); [7]							Urt							
Tassi (1886); [8]						Loa								
Gibson & Warham (1890); [9]										Urt				
Ritterhausen (1892); [10]	Tra													
Giustiniani (1896); [11]					Urt									
Dragendorf (1905); [12]	Urt													
Knoll (1905); [13]									Tra					
Petrie (1906); [14]	Lap					Lap								
Winternitz (1907); [15]	Urt				Urt							Urt		
Flury (1919); [16]	Urt													
Nestler (1925); [17]	Urt						Urt							
Flury (1927); [18]											Urt, Lap			
Kroeber (1928); [19]								Urt						
Starkenstein & Wasserstrom (1933); [20]					Urt			Urt						
Emmelin & Feldberg (1947); [21]		Urt	Urt											
Collier & Chesher (1956); [22]		Urt	Urt	Urt										
Robertson and MacFarlane (1957); [23]		Lap	Lap	Lap										
Pilgrim (1959); [24]														Urt
Saxena et al. (1965, 1966); [25,26]		Urt, Gir	Urt, Gir	Urt, Gir										

Modified from Thurston & Lerston (1969) [1]. Abbreviations: Urt = *Urtica*, Gir = *Girardini*a, Lap = *Laportea* or *Dendrocnide*, Loa = *Loasa,* Tra = *Tragia.*

**Table 2 toxins-13-00141-t002:** Length and lumen (volume) of stinging hairs in various stinging plant species.

Species	Length (mm)	Lumen (nL)
**Urticaceae**		
*Urtica dioica*	1.38	3.85
*Urtica atrovirens*	2.34	13.1
*Urtica mairei*	2.79	38.5
*Urera baccifera*	2	58.5
**Loasaceae**		
*Aosa rupestris*	2.6	10.4
*Caiophora sp.*	2.12	8
*Caiophora deserticola*	3.47	34.4
*Chichicaste grandis*	1.22	4.1
*Loasa pallida*	2.9	19.1
*Nasa amaluzensis*	2.86	31.3
**Euphorbiaceae**		
*Cnidoscolus aconitifolius*	4.25	140.6
**Namaceae**		
*Wigandia spec.*	5.4	44.4

Calculated from transmission light microscopy images of typical stinging hairs from plants grown in the Botanical Gardens, Bonn, Germany.

**Table 3 toxins-13-00141-t003:** Chemical elements including inorganic components in stinging hair fluids.

Plant Family	Species	C	P	Cl	Na	K	S
**Urticaceae**							
	*Laportea perrieri*	100	11	0	0	1.8	2.8
	*Urera baccifera*	110	0	30.4	0	100	0
	*“*	257	4.3	0	0	100	0
	*“*	189	42	0	0	100	0
	*Urtica dioica*	1133	110	0	0	100	10.4
	*“*	791	70	15.2	0	100	15.2
	*“*	167	42	11.1	0	100	0
	*“*	370	173	14	0	100	9.4
	*“*	663	128	18.8	0	100	15
	*Urtica gracilis*	735	97	0	0	100	19
	*Urtica massaica*	600	114	23	128	100	43
	*“*	218	77	15	22	100	22
	*Urtica pilulifera*	280	0	0	0	100	0
**Loasaceae**							
	*Aosa rupestris*	747	0	0	132	100	0
	*Blumenbachia hieronymi*	300	0	0	0	100	0
	*Caiophora lateritia*	467	45	40	38	100	0
	*Loasa elongata*	657	0	0	0	100	0
	*Loasa insons*	294	0	16	0	100	0
	*Loasa pallida*	249	4.8	7.1	0	100	0
	*“*	485	0	20	0	100	0
	*“*	166	0	23	0	100	0
	*Loasa tricolor*	279	17	0	0	100	0
	*Loasa triloba*	162	0	17	3	100	0
	*Nasa amaluzensis*	150	0	0	0	100	0
	*“*	309	0	0	0	100	0
	*Nasa poissoniana*	393	0	0	0	100	0
**Euphorbiaceae**							
	*Cnidoscolus aconitifolius*	177	0	0	0	100	0
	*“*	170	0	0	0	100	0
**Namaceae**							
	*Wigandia caracasana*	101	0	0	0	100	0

Chemical elements in dry mass of stinging hair fluid, measured by EDX. Comparison of the peak height in the spectra, in relation to the potassium (K) peak (=100), except for *Laportea perrieri* due to its very low K content. Each line represents the spectrum of a droplet from a single stinging hair; multiple measurements of the same species illustrate the variability. Replicate measurements from different individual stinging hairs indicate by “.

## Data Availability

Plants used in this study are cultivated at Bonn University Botanic Gardens, vouchers are deposited in the herbarium BONN and are available upon request. Two species *Dalechampia aristolochiifolia* (IPEN XX-0-B-2140101) and *Horovitzia cnidoscoloides* (IPEN MEX-0-M-2013/1841) are in cultivation at Botanischer Garten München-Nymphenburg.

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
