# Peer review of "Distribution, Ecology, Chemistry and Toxicology of Plant Stinging Hairs"

_toxins, 2021, doi:10.3390/toxins13020141_

Round 1

Reviewer 1 Report

This is a very interesting and well prepared review of plants with stinging hairs. For the most part, I don't have any concerns regarding the review. I only have two minor questions.

1- Why did the authors include a materials and methods section in a review article? If this is a review article, there is no need for this section. However, if there is primary data that has not been published previously, which would warrant a description of the materials and methods used to obtain the data, then that needs to be clearly outlined in the manuscript. I was under the assumption that is all review material.

2- The authors state that much of the plant material studied for analysis was obtained from botanical gardens. Is there any data to show that the chemical composition of these stinging hairs, and leaves, are the same, or different, in garden grown plants versus wild plants? Many wild plants have a different chemical profile than that of garden grown plants. If the plants obtained from the botanical garden are different than wild plants, then the analyses are meaningless. If they are the same, then botanical gardens are a good source of obtaining the plant material for analyses.

Author Response

Thank you very much for your constructive cirticism. I hope I can accomodate/refute the two points raised:

1- Why did the authors include a materials and methods section in a review article? If this is a review article, there is no need for this section. However, if there is primary data that has not been published previously, which would warrant a description of the materials and methods used to obtain the data, then that needs to be clearly outlined in the manuscript. I was under the assumption that is all review material.

That is a valid point, but we believe that we are making a good compromise here: We took the liberty to use original images for the review - and therefore had to add the corresponding materials and methods section. Also, we added some original data sets into the paper to update and provide context and an outlook for the published data we review. In order to clarify this point, we now added the following sentence to the end of the ifrst chapter (1. Introduction):

We provide some original data and images (and a corresponding Materials & Methods section) for up-to-date original illustrations and in order to put published work in the context of our own ongoing research. 

2- The authors state that much of the plant material studied for analysis was obtained from botanical gardens. Is there any data to show that the chemical composition of these stinging hairs, and leaves, are the same, or different, in garden grown plants versus wild plants? Many wild plants have a different chemical profile than that of garden grown plants. If the plants obtained from the botanical garden are different than wild plants, then the analyses are meaningless. If they are the same, then botanical gardens are a good source of obtaining the plant material for analyses.

This may, or may not be a valid point, but to date no research has addressed this aspect. Indeed, from cultivating ca. 200 accessions of stinging plants over the past 10 years I (MW) know for certain that there is no perceptible change in the stinging effect of the corresponding species between wild collection and cultivation. Whereever we did address a possible domestication effect in the past (iridoid content, biomineralization) we found no effect. I would wager, that this is also the case here - until somebody prooves the contrary.

Reviewer 2 Report

The singularities of the plants arouse an enormous curiosity in us, their understandings sharpen our perseverance, in a path of advances and retreats. After a period in which the scientific community focused on the morphological and ethnobotanical characterization of plant stinging hairs, with technology evolution the onus change on physiological and metabolomic characterization.

The work present by the author's XXX clearly approaches the subject, not only gathering a set of information from the beginning of studies but also the most recent works. In particular the morphological and metabolomic characterization.

Although recently two revisions, in something complementary (De Vico 2018; Grauso 2020), have appeared - the authors with this publication address some flaws and inaccuracies of previous revision works, due to the number of species they address as well as the diversity of works.

However, even though they have presented an exhaustive work, the authors fail to explore the genetic exponent of the theme (some genes related to hair/functionality (Xu et al., 2019; Kato et al, 2017). A subject that could enrich the present work and embrace a more vast audience.

Congratulations to the authors for the very fine work presented.

Author Response

Thank you very much for your very positive review. You raise one important point: However, even though they have presented an exhaustive work, the authors fail to explore the genetic exponent of the theme (some genes related to hair/functionality (Xu et al., 2019; Kato et al, 2017). A subject that could enrich the present work and embrace a more vast audience.

I fully agree a genetic and metabolomic understanding of the topic would be genuinly interesting. Unfortunately, we are unaware of any directly pertinent studies: We do, however, cite Kato with two of her papers looking at the evolutionary dynamics. The Xu et al. paper looks into the genetics of glandular trichomes - secretory structures that are inherently unlikely to show any parallels in their genetics or physiology. Also, since the title of the volume is "Behavioural Ecology of Venom" we find that looking into the toxins themselves and their distribution and chemistry and ecological relevance are "on topic", but speculating too much about the genetics behind them would be "off topic".